# Longevity and mortality in cats: A single institution necropsy study of 3108 cases (1989–2019)

**Michael S. Kent**[1]*, **Sophie Karchemskiy**[2¤a], **William T. N. Culp**[1], **Amandine T. Lejeune**[1], **Patricia A. Pesavento**[3], **Christine Toedebusch**[1], **Rachel Brady**[2¤b], **Robert Rebhun**[1]

**1** Department of Surgical and Radiological Sciences, School of Veterinary Medicine, University of California, Davis, Davis, CA, United States of America, **2** William T. Prichard Veterinary Medical Teaching Hospital, School of Veterinary Medicine, University of California, Davis, Davis, CA, United States of America, **3** Department of Pathology, Microbiology and Immunology, School of Veterinary Medicine, University of California, Davis, Davis, CA, United States of America

¤a Current address: Animal Medical Center, New York, NY, United States of America
¤b Current address: Cell and Molecular Biology Program, Colorado State University, Fort Collins, CO, United States of America
* mskent@ucdavis.edu

**Data Availability Statement:** All relevant data are within the paper and its Supporting Information files.

## Abstract

Client-owned cats who underwent a post-mortem examination (n = 3,108) at a veterinary medical teaching hospital between 1989 and 2019 were studied to determine longevity and factors affecting mortality. Demographic factors, environmental factors, age, and causes of death were assessed. Sexes included 5.66% intact females, 39.86% spayed females, 6.95% intact males and 47.49% neutered males. 84.2% were mixed breed cats. Age at death was known for 2,974 cases with a median of 9.07 years. Cancer was the most common pathophysiologic cause of death (35.81%) and was identified in 41.3% of cats. When categorized by organ system, mortality was most attributed to multiorgan/systemic (21.72%). Renal histologic abnormalities were noted in 62.84% of cats but was considered the primary cause of death in only 13.06% of cats. Intact female and male cats had significantly shorter lifespans than their spayed or neutered counterparts. FeLV positive status was associated with decreased longevity (P<0.0001) while FIV status was not. This study reports on risk factors associated with mortality and highlights areas of research that may contribute to improved lifespan in cats.

## Introduction

The process of aging and longevity is complex and highly variable among species. Studies designed to identify risk factors, or aging signatures, target multiple levels from molecular, genetics, behavior, ecology. Fundamental to any consideration of longevity in a single species is an understanding of the degenerative, infectious, and non-infectious causes of death.

There has been much recent interest in studies examining longevity and aging in domestic pets as they share their environment with the human population, have access to routine

**Funding:** Center for Companion Animal Health, University of California Davis School of Veterinary Medicine. The funders had no role in study design, data collection and analysis, decision to publish, or preparation of the manuscript.

**Competing interests:** The authors have declared that no competing interests exist.

healthcare including prevention and treatment of disease. While common causes of mortality in dogs have recently been reported, comparatively little is known about mortality and longevity in domestic cats. Cats are one of the most common companion animals in the United States (US). While exact numbers are unknown, it is estimated that as many as 45 million US households have an average of 2 cats each, yielding a total of 90 million owned cats, although another survey indicated a smaller number of households at 37 million and a smaller population of cats in the US at 61 million [1, 2]. Additionally, there are an estimated 30 to 100 million unowned cats living in the US [3–5]. Several studies have looked at causes of death and to some degree factors that affect longevity in owned cats, but this topic remains largely unexplored [6–8].

Several studies have looked at individual risk factors for their impact on longevity including cardiovascular disease and body condition score, or specifically evaluated factors that affected survival in cats who tested positive for feline leukemia virus (FeLV) or feline immunodeficiency virus (FIV) infection. Large scale studies assessing overall longevity in cats and the causes of mortality are limited [6–8]. A 2015 study looked at a primary care database of cats in the UK and studied over 4,000 cats who had died [9]. That study determined the median longevity of cats to be 14.0 years with mixed breed cats living longer than pure bred cats. In that study, 91.7% were mixed breed cats, with just over half of the cats being female (50.7%) and the majority of cats being spayed or neutered (64.8%). The most frequently attributed causes of mortality in cats of all ages were trauma (12.2%), renal disorder (12.1%), non-specific illness (11.2%), cancer (10.8%) and mass lesion disorders (10.2%). Increased longevity was associated with being crossbred, having a lower bodyweight and being neutered. However, the cause of death was entered by the attending clinician, and it is not known whether these diagnoses were confirmed by biopsy or histopathology. Necropsy examinations can provide information as to the cause of death with greater accuracy than clinical diagnoses alone can. Further, in dogs it has been shown that clinical diagnoses were not consistently confirmed by necropsy, which often reveals alternate causes of death, or complex processes not understood without necropsy [10, 11].

We hypothesized that evaluation of a historical set of necropsy examinations combined with clinical annotations will provide information on longevity, identify causes of mortality and highlight risk factors affecting survival in cats in the US. We aimed to analyze this data by classifying death organ system and pathophysiological process to determine which mortally determining disease processes are most common in cats. By collecting information such as age of death and other factors including signalment, viral status, housing information, the presence of cancer, heart disease, and renal disease we also aimed to determine factors that may impact longevity.

## Materials and methods

### Case selection

The electronic medical record database of the UC Davis William R. Pritchard Veterinary Medical Teaching Hospital was searched for all cats undergoing necropsy examinations between January 1, 1989 and October 31, 2019. Client consent for all necropsy examinations were obtained verbally by the attending clinician or through a signed consent form, with the understanding that resulting data and collected tissues could be used for further research. Client-owned cats with a complete necropsy report were included in the study. Cats were excluded if they belonged to shelters, rescue or fostering organizations, research colonies, or were otherwise unowned or had unidentified owners. Client-owned cats who presented dead on arrival to the hospital without having been previous patients, and those submitted only for rabies

testing were likewise excluded. Client-owned cats who were surrendered to the hospital prior to their death were included.

The following demographic information was collected for each cat included in the study: birthdate or estimated birthdate, date that necropsy was requested, sex, reproductive status, breed, coat color, name, and body weight. Method of death was classified as humane euthanasia vs. natural. Breeds were reported by the owner and classified according to The International Cat Association's list of breeds. For those cats with multiple body weights recorded, the last reported weight or the most consistently reported weight was used. Body weights were assumed to be measured in kilograms if not otherwise stated, and estimated weights were not included. In some cases, weight was calculated based on an organ's percentage of body weight reported at necropsy. Those classified as having been euthanized did not include those who were not resuscitated post-cardiac or respiratory arrest, regardless of whether they received a euthanasia injection. Patients who were placed on a ventilator post respiratory arrest prior to euthanasia were categorized as having been euthanized.

In addition to the above demographics, cats were also classified as "indoor only," "outdoor only," or "indoor-outdoor." "Indoor only" cats were explicitly described as kept inside and included those with access to a balcony or those who escaped outdoors for a short time despite the clients' intention to keep them indoors. "Outdoor only" cats were comprised of those explicitly identified as outdoor, including those with access to a barn or garage. "Indoor-outdoor" cats included those described as such, as well as indoor cats who escaped outside occasionally or for prolonged periods of time, indoor cats who had access to an outdoor enclosure or barn, and those who were primarily indoor or primarily outdoor. Otherwise, cats were classified as "unknown." When this classification changed over a patient's lifetime or was described inconsistently, the status with the greatest exposure to the outdoors reported within a year of death was recorded. If it was not known when this change occurred, the cats were categorized by their highest outdoor exposure.

The retroviral status was also collected from the medical record when available, specifically for feline leukemia virus (FeLV) and feline immunodeficiency virus (FIV). If a cat tested positive for FeLV or FIV on any test, such as ELISA antigen or antibody testing, immunohistochemistry, or polymerase chain reaction, the cat was classified as FeLV or FIV positive. Weakly positive test results were recorded as positive. If a previously FeLV positive cat reverted to a negative status, the negative result was recorded. Client-reported results were acceptable. When the retroviral status was not reported, it was classified as "unknown."

The primary cause of death, main organ system affected, and pathophysiological process were assigned for each cat by one of the authors (MSK). The cause of death was determined based on necropsy findings. When the primary cause could not be established, it was recorded as undetermined. The organ system and pathophysiological process involved in the cause of death were classified based on a scheme modified from a previous study on dogs [12]. The 15 organ system classifications included: behavioral, cardiovascular, dermatologic, endocrine, gastrointestinal, genital, hematopoietic, hepato-biliary, musculoskeletal, neurologic, ophthalmologic, respiratory, urological, multiorgan/systemic, and undetermined. The multiorgan/systemic category included cases in which multiple organs were affected, with diseases such as non-localized lymphoma and feline infectious peritonitis or trauma affecting multiple organ systems. The 10 pathophysiological classifications were comprised of the following: congenital (including anomalous), degenerative, infectious, inflammatory (including immune-mediated), ischemic, metabolic, neoplastic, toxic, traumatic, vascular, and undetermined. The following findings were also recorded if present: cancer, renal and cardiac changes, and hyperthyroidism; the latter was defined as either a clinical history of hyperthyroidism or the presence of a thyroid adenoma or thyroid hyperplasia. The histologic diagnosis for cancer and renal disease

was recorded, as well as whether the cancer or renal and cardiac changes led to a cat's death. Renal changes were categorized by histological classifications including tubular/interstitial disease, glomerular disease, changes consistent with toxin exposure, end-stage renal disease, cancer, pyelonephritis, infarcts, polycystic kidney disease, changes consistent with feline infectious peritonitis, papillary necrosis, amyloidosis, the presence of renal calculi or nephroliths, tubular necrosis, congenital hypoplasia or aplasia and a miscellaneous category for those diagnoses that had one or two cases only. In cases where there were questions on wording of the interpretation or it was unclear in the necropsy report, a single boarded veterinary pathologist reviewed the stored histology samples to allow for classification. Tumors were categorized as being of epithelial, mesenchymal or round cell origin. A separate category was used for benign tumors, such as pituitary macroadenomas, that led to death or euthanasia for instance due to tissue compression, hemorrhage, quality of life, or other.

## Statistical analyses

Data was recorded in a commercially available spreadsheet, and statistical analyses were conducted using a commercially available statistics program (Stata version 14.2, Stata Corporation, College Station Texas, USA). Descriptive statistics were performed to report demographic data, such as survival times, sex, and indoor/outdoor status, and to evaluate the frequency of FeLV or FIV infections, causes of death, cancer, renal and cardiac changes, and thyroid changes consistent with hyperthyroidism. Continuous data was assessed for normality by visualization of distributional plots and use of a Shapiro-Francia normality test. When continuous data was normally distributed, means and standard deviations were reported; otherwise, medians, interquartile ranges, and overall range were reported. Totals and percentages were used to describe categorical data. Differences in median survival times between categorical groups were assessed using either a t-test or ANOVA or a Mann-Whitney or Kruskal-Wallis test, depending on data normality and the number of categorical variables. For categorical variables where there were more than two possible outcomes, a Dunn's test with Bonferroni corrections were done to look for differences between groups which were corrected for multiple comparisons. Analysis was done on all cats and then also for cats $\geq$ 1 year. Associations between categorical data were evaluated using a chi squared or Fisher's exact test. To explore the effects of the categorical variables on age, regression analysis was performed. P values <0.05 were considered significant.

## Results

A total of 3,511 cats were available for this study. A total of 3,108 cases with owner consent for complete necropsy were included and 403 cases were excluded based on the following reasons: 296 of the cats were not client-owned or an owner could not be identified; 75 cats were brought to the hospital already deceased and there was no previous patient-doctor relationship with any clinician at the hospital; 29 were research animals; 2 cats were presented to the pathology service for rabies testing only and a full necropsy examination was not performed; and one case in which a necropsy exam was requested, but the results were not recorded in the electronic medical record and could not be located. The dataset used in this study is available as S1 Dataset. The median number of necropsy examinations per year was 100 (range 50–137). The median number of individual cats seen each year at the hospital was 2288 (range 1870–3133). The median percentage of cats undergoing necropsy examination in comparison to the number of cats seen each year was 4.59% (range 1.81–6.16%) which significantly decreased over time during the study period, $R^2$ 0.57, F(1,29)– 38.23, P<0.0001 (S1 Fig).

## Demographics

Included in the study were 176 (5.66%) intact females, 1,239 (39.86%) spayed females, 216 (6.95%) intact males and 1,476 (47.49%) neutered males. In one case, the sex was not recorded. There were 2,618 (84.2%) mixed breed and 490 (15.8%) purebred cats included in the study. A total of 26 different breeds of cats were examined. For 16 cats who were reported as purebred, their exact breed could not be defined according to the chosen classification system. For a complete breakdown of cat breeds see S1 Table. The median weight, available for 2,962 cats (%), was 4.0Kg (range 0.05–15.45Kg).

## Age at time of death

**Effect of method of death.**  In total, 512 (16.5%) cats died, and 2,596 (83.5%) cats were euthanized. Of the 2,974 cats with documented ages, the median age at death was 9.07 years (IQR 4.20–12.92 years; range 0.01–21.85 years). The median age of death for cats who died naturally was 8.27 years (IQR 3.84–12.25 years; range 0.01–21.24 years), while the median age at death for euthanized cats was 9.21 years (IQR 4.24–12.03 years; range 0.01–21.85 years). These were significantly different (p = 0.02). A total of 2,672 (89.85%) cats were ≥1 years of age at the time of death. The median overall survival for this group was 9.92 years (IQR 6–13.25; range 1.00–21.85 years). Age was not found to be normally distributed when looking at all cats or in the group of cats ≥ 1year of age at the time of death. A histogram of the age at death for both groups is shown in Fig 1.

## Environmental, sex and viral infection status

Information regarding where the cat lived was available for 2,284 cats (73.49%). Of all cats, 1,023 (32.9%) cats were reported to live indoor only, 1,071 (34.5%) lived both indoor and outdoor and 190 (34.5%) lived outdoors exclusively. To determine if there was a relationship between sex of the cat and their environment, we further analyzed housing by sex. Of the intact female cats, 38 (42.22%) were indoor only, 40 (44.44%) were indoor outdoor and 12 (13.33%) were outdoor only. Amongst female spayed cats, 458 (49.09%) were indoor only, 411 (44.05%)

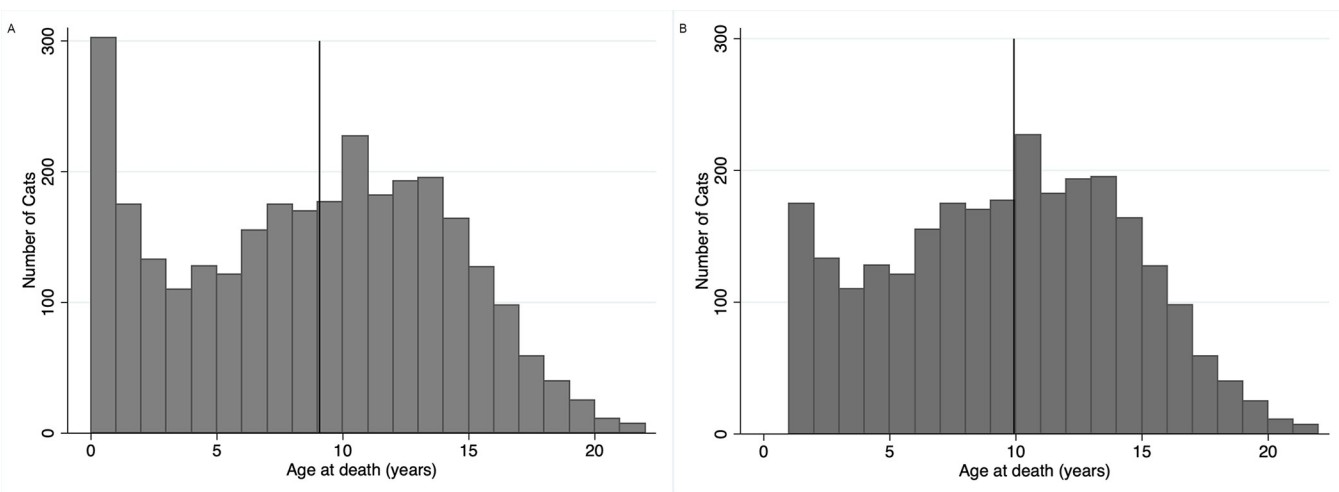

**Fig 1.** Distribution of age at death for cats (A) Histogram showing the ages of death for the 2,974 cats in the study for which their age was known. The line represents the median age of death at 9.07 years. (B) Histogram showing the ages of death for the 2,672 cats in the study for which their age was known and they were ≥ 1 year. The line represents the median age of death at 9.92 years.

were indoor outdoor and 64 (6.86%) were outdoor only. Within female cats, there was no difference in the distribution of where they lived between intact and spayed cats (p = 0.07).

For intact male cats, 47 (34.56%) were indoor only, 54 (39.71%) were indoor outdoor and 35 (25.74%) were outdoor only. For castrated male cats, 480 (42.67%) were indoor only, 566 (50.31%) were indoor outdoor and 79 (7.02) were outdoor only. The proportion of intact versus castrated male cats living in different environments was significantly different (p<0.0001).

To explore the possibility that cats with an outdoor exposure were at increased risk for developing potentially important infectious disease FeLV status and environment was looked at. The FeLV status was known for 1,626 (52.32%) of the cats, with 110 (3.5%) testing positive. Of the FeLV positive cats, 101 had an age of death available, with 89 ≥ 1 year of age. FIV status was known for 1,402 cats (45.11%), with 84 (5.99%) testing positive. Of the FIV positive cats, 78 had an age of death available, with 76 ≥1 year of age. Nine cats tested positive for both FeLV and FIV. 41 cats who tested positive for FeLV did not have a recorded FIV test, and 18 cats who tested positive for FIV did not have a recorded FeLV test. Intact male cats had a higher proportion of positive FeLV testing (15/21 (12.4%) compared to neutered males (51/ 811 (6.29%; p = 0.02). Three of the 81 (3.7%) intact females were FeLV positive while 41 of the 613 (6.69%) spayed female cats were FeLV positive; this difference was not statistically different (p = 0.22). Moreover, there was no effect of sex on the proportion of FIV positive cats. Eight of 94 (8.51%) intact males were FIV positive and 58 of 714 (8.12%) of neutered males were FIV positive (p = 0.09). Four of 64 (6.25%) intact female cats were FIV positive, while 14 of 530 (2.64%) spayed female cats were FIV (p = 0.12).

A total of 1,296 cats had records of both their FeLV status and housing situation. Of the 592 cats that were indoor only, 27 (4.56%) were positive for FeLV infection. Of the 106 outdoor only cats for which FeLV status was known, 15 (14.15%) were positive, and for the 598 cats whose housing situation was indoor/outdoor, 36 (6.02%) were positive. These groups were statistically different overall (p = 0.002) with indoor only cats and cats that lived indoor/outdoor being having a lower proportion of infected cats then outdoor only cats (p = 0.0001 and p = 0.0006 respectively. There was no difference between cats that lived indoor only and those that lived indoor/outdoor (p = 0.15).

Similarly, 1,133 cats had records of their FIV status and housing situation. Of the 530 indoor only cats, 25 (4.72%) were positive for FIV, while 9 of 91 outdoor only cats, (9.90%) were positive for FIV. For the 512 cats who lived indoor/outdoor, 28 (5.47%) tested positive for FIV. These numbers were not statistically different (p = 0.13).

## Causes of mortality

The most common organ system identified as the driver of mortality was the multiorgan/systemic category (21.72%), followed by the urological (13.93%), respiratory (11.42%), neurological (10.62%) and gastrointestinal systems (10.04%). In 45 (1.45%) cats, a primary organ system (s) could not be identified as the cause of death. The most common pathophysiologic cause of death was cancer (35.81%) followed by infectious causes (17.47%) and degenerative causes (11.97%). In 322 (10.36%) cases the main pathophysiologic cause of death was not identified. A complete list of the categorized causes of death by organ system and pathophysiology are presented in Tables 1 and 2, respectively.

Causes of death were further classified by disease process. The most common cause of death was cancer with 1,113 cases (35.81%). This was followed by renal failure, feline infectious peritonitis, and cardiac disease, with these processes leading to death in 336 (10.81%), 209 (6.72%) and 161 (5.18%) cats, respectively. These four causes of death accounted for 58.52% percent of all deaths. A complete listing of all causes of death for the cats in this study are listed in S2 Table.

**Table 1. Cause of death by organ system, based on necropsy findings, for all cats and for cats ≥ 1 year of age.**

| Organ system cause of death | All Cats | | | | | | Cats ≥1 year | | | |
|---|---|---|---|---|---|---|---|---|---|---|
| | Number | Percent | Number with Age | Median Age (Years) | IQR (Years) | Range (Years) | Number with Age | Median Age (Years) | IQR (Years) | Range (Years) |
| Cardiovascular | 299 | 9.62 | 280 | 8.41 | 4.31–12.02 | 0.04–21.13 | 264 | 8.73 | 5.00–12.15 | 1.00–21.13 |
| Dermatologic | 76 | 2.45 | 73 | 10.39 | 6.26–13.55 | .51–18.41 | 69 | 10.44 | 7.77–13.61 | 1.53–18.41 |
| Endocrine | 173 | 5.57 | 165 | 12.65 | 9.67–15.01 | 1.02–21.85 | 165 | 12.65 | 9.67–15.01 | 1.02–21.85 |
| Gastrointestinal | 312 | 10.04 | 298 | 10.51 | 7.03–13.66 | 0.08–21.24 | 277 | 10.85 | 7.96–13.83 | 1.11–21.24 |
| Genital | 22 | 0.71 | 21 | 9.55 | 4.96–12.45 | 0.24–19.87 | 20 | 9.93 | 5.48–12.74 | 1.05–19.87 |
| Hematopoetic | 105 | 3.38 | 104 | 7.04 | 2.52–11.07 | 0.22–19.90 | 91 | 8.39 | 4.00–11.22 | 1.05–19.91 |
| Hepato-billiary | 158 | 5.08 | 151 | 8.52 | 5.48–11.72 | 0.02–21.44 | 146 | 8.67 | 6.02–11.81 | 1.35–21.44 |
| Musculoskeletal | 119 | 3.83 | 114 | 8.69 | 4.79–12.11 | 0.01–19.39 | 101 | 9.07 | 6.80–12.50 | 1.08–19.39 |
| Neurological | 330 | 10.62 | 324 | 8.24 | 3.66–12.66 | 0.06–20.66 | 290 | 9.07 | 5.16–13.02 | 1.00–20.66 |
| Ophthalmologic | 6 | 0.19 | 6 | 11.45 | 9.32–16.17 | 0.26–18.07 | 5 | 11.76 | 11.13–16.18 | 9.32–18.07 |
| Respiratory | 355 | 11.42 | 343 | 9.87 | 5.71–13.13 | 0.01–21.12 | 317 | 10.32 | 6.77–13.33 | 1.11–20.12 |
| Systemic | 675 | 21.72 | 643 | 6.25 | 1.38–11.53 | 0.05–20.44 | 508 | 8.75 | 4.00–12.39 | 1.01–20.44 |

**Table 2. Cause of death by pathophysiologic process, based on necropsy findings, for all cats and for cats ≥ 1 year of age.**

| Pathophysiologic cause of death | | | All Cats | | | | Cats ≥1 year | | | |
|---|---|---|---|---|---|---|---|---|---|---|
| | Number | Percent | Number with Age | Median Age (Years) | IQR (Years) | Range (Years) | Number with Age | Median Age (Years) | IQR (Years) | Range (Years) |
| Anatomic/congenital | 85 | 2.73 | 80 | 3.75 | 0.79–7.63 | 0.01–21.24 | 58 | 5.17 | 3.07–9.63 | 1.01–21.24 |
| Degenerative | 372 | 11.97 | 350 | 10.48 | 6.71–14.01 | 0.25–21.75 | 344 | 10.65 | 6.81–14.05 | 1.16–21.75 |
| Infectious | 543 | 17.47 | 523 | 3.45 | 0.79–8.64 | 0.01–19.85 | 362 | 6.26 | 3.04–10.64 | 1.01–19.85 |
| Inflammatory | 205 | 6.6 | 199 | 8.67 | 4.21–12.16 | 0.12–17.61 | 186 | 9.17 | 5.03–12.31 | 1.52–17.61 |
| Ischemic | 122 | 3.93 | 117 | 7.93 | 4.44–11.68 | 0.12–21.13 | 112 | 8.18 | 4.92–11.95 | 1.11–21.13 |
| Metabolic | 130 | 4.18 | 125 | 9.62 | 6.58–13.21 | 0.36–21.45 | 122 | 9.86 | 6.68–13.22 | 1.13–21.44 |
| Neoplastic | 1,113 | 35.81 | 1069 | 11.43 | 8.60–14.02 | 0.22–21.85 | 1054 | 11.58 | 8.76–14.04 | 1.01–21.85 |
| Nutritional | 2 | 0.06 | 2 | 2.49 | 0.53–4.44 | 0.53–4.44 | 1 | 4.44 | NA | 4.44 |
| Toxic | 69 | 2.22 | 66 | 5.01 | 1.95–8.01 | 0.51–16.58 | 58 | 5.72 | 2.69–10.03 | 1.00–16.58 |
| Traumatic | 145 | 4.67 | 130 | 3.35 | 1.03–9.04 | 0.11–18.07 | 99 | 5.60 | 2.16–10.04 | 1.02–18.07 |
| Undetermined | 322 | 10.36 | 313 | 7.35 | 3.23–11.64 | 0.01–20.69 | 276 | 8.00 | 4.55–12.04 | 1.02–20.69 |

**Table 3. Frequency of occurrence of tumor types and number of tumors seen in each cat.**

| | Number of Cats | Mesenchymal | Multiple Mesenchymal | Epithelial | Multiple Epithelial | Round cell | Multiple Round cell | Benign | Unclassified |
|---|---|---|---|---|---|---|---|---|---|
| Mesenchymal | | 177 | NA | 15 | 1 | 23 | 2 | 1 | 1 |
| Multiple Mesenchymal | | NA | 1 | 0 | 0 | 0 | 0 | 1 | 0 |
| Epithelial | | 15 | 0 | 435 | NA | 67 | 4 | 7 | 0 |
| Multiple Epithelial | | 1 | 0 | NA | 21 | 3 | 1 | 0 | 0 |
| Round cell | | 25 | 0 | 67 | 3 | 479 | NA | 4 | 1 |
| Multiple Round | | 2 | 0 | 4 | 1 | NA | 29 | 0 | 0 |
| Benign | | 1 | 0 | 7 | 1 | 5 | 0 | 30 | 0 |
| Unclassified | | 0 | 0 | 0 | 0 | 1 | 0 | 0 | 1 |
| Total Path Diagnosis | | 221 | 3 | 528 | 27 | 578 | 36 | 43 | 3 |
| Total Cats | 1283 | 215 | 1 | 523 | 23 | 573 | 29 | 42 | 2 |
| Cats with one tumor | 1122 | 177 | NA | 435 | NA | 479 | NA | 30 | 1 |
| Cats with two tumors | 146 | 32 | 0 | 80 | 18 | 88 | 24 | 10 | 0 |
| Cats with 3 tumors | 12 | 4 | 1 | 7 | 3 | 7 | 1 | 1 | 1 |
| Cats with 4 Tumors | 2 | 1 | 0 | 1 | 1 | 0 | 1 | 1 | 0 |
| Cats with 5 tumors | 1 | 1 | 0 | 0 | 1 | 0 | 1 | 0 | 0 |

## Cancer

Cancer was identified in 1,283 (41.3%) cats. The identified cancer was the reported cause of death in 1,113 cats which was 35.81% of all cases and 86.75% of cases with cancer. In 170 cats diagnosed with cancer, their cancer was not the primary cause of death which represented 5.47% of all cases and 13.25% of cases with cancer. The number of solid tumors an individual cat was diagnosed with ranged from 0–5. Of the 1,283 cats diagnosed with cancer, 1122 (87.45%) were diagnosed with 1 cancer, 146 (11.38%) were diagnosed with 2 cancers, 12 (0.94%) were diagnosed with 3 cancers, 2 (0.16%) were diagnosed with 4 cancers and 1 (0.08%) was diagnosed with five distinct histologic types of cancer. Amongst the cats diagnosed with a single cancer, the most common type of cancer was round cell tumors (n = 479; 42.69%), followed by epithelial origin tumors (435; 38.77%) and mesenchymal origin tumors (177;15.078%). Thirty (2.67%) tumors were considered benign histologically but led to death, and one tumor was undifferentiated in a manner that prevented the cell of origin from being classified. A complete table of tumor categories broken down by the number of cancers for each cat is available in Table 3.

## Renal disease

A total of 1953 (62.84%) cats were found to have histologic evidence of renal disease on necropsy examination. Renal disease was considered the cause of death in 406 cats, (13.16% of total cases and 20.8% of cases with renal disease), whereas the majority of affected cats (1,547 or 49.8% of all cases and 79.2% of those with renal pathology) died of another cause or causes. Of the cats with renal disease, the majority had a single pathological process (1,517; 77.7%). Multiple renal pathologies were diagnosed in a minority of cats, with two independent renal lesions found in 386 cats (19.8%), three in 46 cats (2.3%), and four in 4 cats (0.2%). Most (1,207, 61.8%) cats with renal disease were classified as having tubular/interstitial disease. In 875 cases (44.8%), tubular/interstitial disease was the only histopathological renal finding while in another 325 cases multiple processes such as glomerular, pelvic or other portions of

**Table 4. Renal disease processes as determined on necropsy examination in cats.**

| | Number of cats | TI | Gl | Toxic | End Stage | Cancer | Pyelo | Misc | Infarct | Poly | FIP | Papillary Necrosis | Am | Calculi, nephroliths | tubular necrosis | Cong |
|---|---|---|---|---|---|---|---|---|---|---|---|---|---|---|---|---|
| TI | | 875 | 7 | 7 | 3 | 22 | 33 | 2 | 163 | 9 | 0 | 32 | 14 | 37 | 41 | 4 |
| GI | | 7 | 43 | 0 | 0 | 1 | 2 | 0 | 3 | 0 | 0 | 0 | 0 | 0 | 0 | 0 |
| Toxic | | 7 | 0 | 32 | 0 | 0 | 0 | 0 | 5 | 0 | 0 | 0 | 0 | 1 | 0 | 0 |
| End Stage | | 3 | 0 | 0 | 131 | 1 | 16 | 1 | 15 | 0 | 0 | 1 | 1 | 10 | 12 | 0 |
| Cancer | | 22 | 1 | 0 | 1 | 64 | 2 | 0 | 10 | 0 | 0 | 0 | 0 | 0 | 2 | 1 |
| Pyelo | | 33 | 2 | 0 | 16 | 2 | 94 | 0 | 15 | 2 | 1 | 8 | 2 | 12 | 5 | 0 |
| Misc | | 2 | 0 | 0 | 1 | 0 | 0 | 32 | 2 | 0 | 0 | 0 | 0 | 0 | 0 | 2 |
| Infarct | | 163 | 3 | 5 | 15 | 10 | 15 | 2 | 109 | 0 | 2 | 8 | 0 | 13 | 8 | 0 |
| Poly | | 9 | 0 | 0 | 0 | 0 | 2 | 0 | 0 | 9 | 0 | 1 | 0 | 0 | 1 | 0 |
| FIP | | 0 | 0 | 0 | 0 | 0 | 1 | 0 | 2 | 0 | 75 | 0 | 0 | 0 | 0 | 0 |
| Papillary Necrosis | | 32 | 0 | 0 | 1 | 0 | 8 | 0 | 8 | 1 | 0 | 5 | 0 | 3 | 2 | 0 |
| Am | | 14 | 0 | 0 | 1 | 0 | 2 | 0 | 0 | 0 | 0 | 0 | 6 | 0 | 0 | 0 |
| Calculi, nephroliths | | 37 | 0 | 1 | 10 | 0 | 12 | 0 | 13 | 0 | 0 | 3 | 0 | 9 | 2 | 0 |
| Tubular necrosis | | 41 | 0 | 0 | 12 | 2 | 5 | 0 | 8 | 1 | 0 | 2 | 0 | 2 | 28 | 1 |
| Cong | | 4 | 0 | 0 | 0 | 1 | 0 | 2 | 0 | 0 | 0 | 0 | 0 | 0 | 1 | 5 |
| Number of pathological diagnosis | | 1249 | 56 | 45 | 191 | 103 | 192 | 39 | 353 | 22 | 78 | 60 | 23 | 87 | 102 | 13 |
| Number of cats | 1953 | 1207 | 55 | 41 | 184 | 98 | 170 | 39 | 317 | 19 | 78 | 46 | 23 | 66 | 88 | 12 |
| Cats with one pathological diagnosis | 1517 | 875 | 43 | 32 | 131 | 64 | 94 | 32 | 109 | 9 | 75 | 5 | 6 | 9 | 28 | 5 |
| Cats with two pathological diagnosis | 386 | 293 | 11 | 6 | 47 | 29 | 56 | 7 | 173 | 8 | 3 | 29 | 17 | 39 | 48 | 6 |
| Cats with three pathological diagnosis | 46 | 36 | 1 | 2 | 5 | 5 | 18 | 0 | 34 | 1 | 0 | 10 | 0 | 15 | 10 | 1 |
| Cats with four pathological diagnosis | 4 | 3 | 0 | 1 | 1 | 0 | 2 | 0 | 1 | 1 | 0 | 2 | 0 | 3 | 2 | 0 |

Table 4 abbreviations: TI = Tubular/Interstitial Disease; GD = Glomerular Disease; Pyleo = Pyelonephritis; Misc = Miscellaneous; Poly = Polycystic; AM = Amyloidosis; Cong = Congenital hypoplasia/aplasia

the nephron in addition to tubular/interstitial disease were identified. For a complete breakdown of kidney disease in these cats see Table 4. This table also shows which cats had multiple renal pathologies and which type of kidney disease occurred together in a single cat.

## Cardiac disease

A total of 867 (27.90%) cats were found to have some form of cardiac pathology with it being attributed as the cause of death in 224 (25.84%) of those cases. The most common cardiac abnormality noted was hypertrophic cardiomyopathy (392 cases with it being identified as the cause of death in 145 cases). This was followed by endocardiosis with 85 cases (where it was identified as the main driver of mortality in 13 cases), myocarditis with 59 cases (where it cause death in 8 cases), and myocardial fibrosis (where it was identified as the cause of death in 8 cases).

## Hyperthyroidism

Hyperthyroidism was diagnosed in 521 of 3108 cats (16.8%). Of these 521 cases, 461 were mixed breed (17.61% of mixed breed cats) and 60 were purebred cats (12.24% of purebred cats); thus, a significantly higher proportion of mixed breed cats had hyperthyroidism (p = 0.004). 11.1% of outdoor only cats were diagnosed with hyperthyroidism which was significantly lower than indoor cats (18.9%) or indoor/outdoor cats (19.1%) diagnosed with hyperthyroidism (p = 0.03).

## Factors affecting longevity

**Effect of sex.** The median age of death for intact female cats was 1.05 years (IQR 0.45–4.73 years; range 0.01–21.15 years) while the median age of death for spayed female cats was 10.28 years (IQR 6.27–13.59 years; range 0.22–21.85 years); these were significantly different (p<0.0001). The median age of death for an intact male cat was 0.79 years (IQR 0.38–3.08 years; range 0.01–19.05 years), while the median age of death for castrated males was 9.55 years (IQR 5.30–12.83 years; range 0.18–21.19 years). These differences were also significantly different (p = 0.0001). The median lifespan of intact female cats ≥ 1 year of age was 4.68 years (IQR 2.03–10.36 years; range 1.01–21.15 years) while the median lifespan of spayed females in this age group was 10.48 years (IQR 6.97–13.68; range 1.01–21.85 years). These were significantly different (P = 0.0001). Of those who died at ≥ 1 year of age, the median lifespan of intact male cats was 3.67 years (IQR 1.96–8.70; range 1.01–19.05 years) while the median lifespan of male neutered cats was 9.84 years (IQR 6.06–13.04 years). These were significantly different (P = 0.0001), with data presented in Fig 2.

**Effect of environment/housing.** The median age at death for indoor only cats was 9.43 years (IQR 4.8–13.11 years, range 0.11–21.85 years) while the median age at death for indoor outdoor cats was 9.82 years (IQR 5.3–13.13 years, range 0.06–21.19 years) and the median age for outdoor cats was 7.25 years (IQR 1.78–11.92 years, range 0.12–20.64 years). These were statistically different (p = 0.0001) with outdoor cats having a shorter lifespan than either indoor only cats (p = 0.0001) or cats that lived indoor/outdoor (p<0.0001). There was no difference in the age of death between indoor only cats and those that lived indoor/outdoor. For cats ≥1

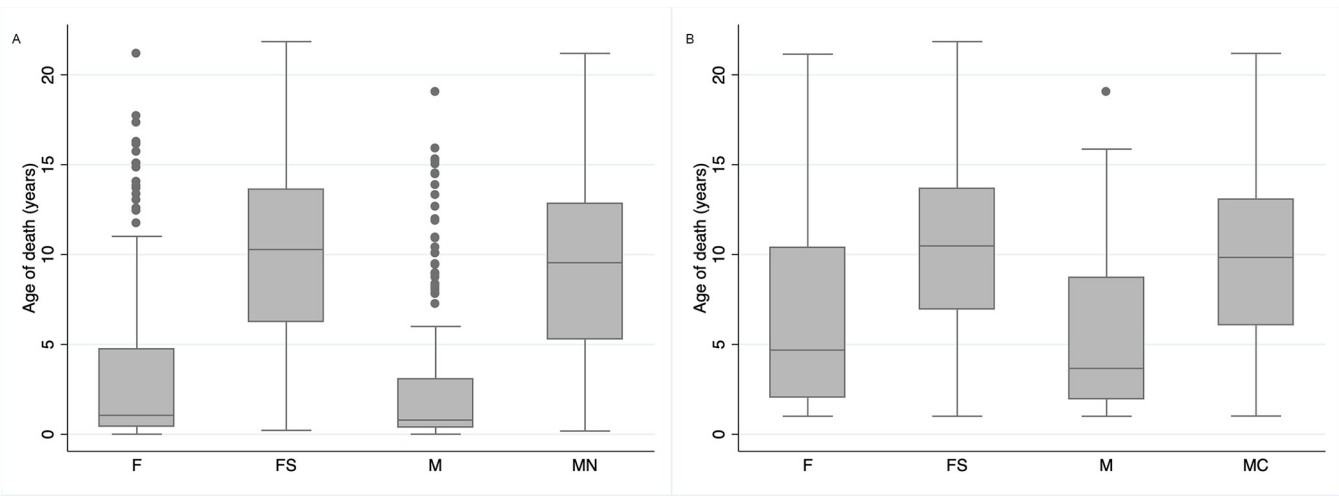

**Fig 2. Spay and neuter status affects longevity in cats.** (A) Box plot showing age at time of death for all cats where age was known categorized by sex. (B) Box plot showing age at time of death for all cats ≥ 1 year where age was known categorized by sex. F = Intact Female, FS = Female spayed, M = Intact male and MN = male neutered.

year of age, the median age of death for indoor cats was 9.98 years (IQR 6.14–13.46 years, range 1.01–21.85 years) while the median age of death for indoor outdoor cats was 10.09 years (IQR 6.29–13.35 years; range 1.00–21.19 years) and the median age of death for outdoor cats was 9.80 years (IQR 4.07–12.92 years). These differences were not statistically different (p = 0.11).

**Effect of FeLV/FIV infection.** The median age of death for an FeLV positive cat amongst cats of all ages was 3.89 years (IQR 1.64–7.58 years; range 0.22–15.25 years) while the median age of death for an FeLV negative cat was 8.70 years (IQR 4.03–12.17 years; range 0.12–21.85 years). These ages were significantly different (P<0.0001). When examining cats ≥1 year of age (N = 89), the median age of death for an FeLV positive cat was 4.28 years (IQR 2.35–8.61 years; range 1.05–15.25 years) while for FeLV negative cats >1 year of age was 9.42 years (IQR 5.83–12.65 years; range 1.01–21.85 years), which were significantly different (P = 0.0001). The median age of death for FIV positive cats was 9.19 years (IQR 6.45–13.15 years; range 0.22–15.25 years), while the median age of death for an FIV negative cat was 8.65 years (IQR 4.16–12.15 years; range 0.35–20.44 years). Unlike FeLV, FIV status did not affect median survival time when all cats were included (p = 0.08). For cats ≥ 1 year of age, the median age at death for an FIV positive cat was 9.38 (IQR 6.70–13.40; range 1.22–20.44) while the median age of death for an FIV negative cat was 9.26 years (IQR 5.59–12.55 years; range 1.01–21.85 years). These ages were also not statistically different (p = 0.42).

**Multivariate analysis of demographics.** To explore the effects that the cat's environment, breed, FeLV and FIV status, sex and spay/neuter status had on age of death, linear regression analysis was done. The overall model was significant at F (8,1067) = 17.80, p<0.0001 with an $R^2$ = 0.12. Being FeLV positive (p<0.001) and being intact (p<0.0001) were associated with a decreased survival time. However, being a spayed female (p<0.0001) and being a neutered male (p<0.0001) were associated with an increased survival time. Environmental status, FIV status, and breed status did not have significant effects on the model.

**Organ system and pathophysiologic effect on cause of death.** Age at the time of death was calculated for each of the different organ systems for all cats and for those cats ≥ 1 year of age. Those with endocrine organ disease had the highest median age at death at 12.65 (IQR 9.67–15.01 years, range 1.02–21.85 years) years for all cats. The data for all organ systems is presented in Table 1. Age at time of death was also calculated for each pathophysiologic process for all cats and for cats ≥ 1 year of age. Cats dying of cancer were the oldest with a median age of death being 11.43 (IQR 8.60–14.02 years; range 0.22–21.85 years) years for all cats and 11.58 (IQR 8.76–14.04 years; range 1.01–21.85 years) years for cats ≥ 1 year of age. The data for all pathophysiologic processes are presented in Table 2.

There were a total of 1914 cats with histologic evidence of renal disease. The median survival overall was 10.35 years (IQR 6.28–13.72 years; range 0.01–21.85 years). Cats who died of their renal disease had a median age of death of 9.41 (IQR 5.50–13.46 years; range 0.11–21.75 years) while those who were classified with less significant renal disease and died of another cause had a median age at death of 10.52 years (IQR 6.61–13.78 years; range 0.01–21.85 years). These ages of death were statistically different (p<0.01). There were 832 cats who had signs of cardiac pathology on their necropsy examination. Cats who died of cardiac disease had a median age of death at 8.42 years (IQR 4.68–12.07 years; range 0.10–21.13 years), while those who had cardiac pathology found but died of other causes had a median age of death at 10.84 years (IQR 6.01–14.02 years; range 0.11–21.75 years). The age at death was statistically different between the two groups (p<0.0001). Those cats who had no signs of cardiac disease on their necropsy lived a median of 8.70 years (IQR 3.56–12.59 years; range 0.01–21.85 years). Of cats with a known age of death diagnosed with cancer (n = 1914), the median age of death was 11.75 years (IQR 8.78–14.31 years; range 0.22–21.85 years). Cats who died of their cancer had

a median age of death of 11.46 years (IQR 8.61–14.04 years; range 0.22–21.85 years), cats who were diagnosed with cancer but died of other causes had a median age at death of 13.13 years (IQR 10.19–15.76 years, range 1.11–21.75 years). These ages were statistically different (p<0.001).

**Effect of spay or neuter status on longevity by cause of death.** Intact reproductive status was found to result in a decreased longevity, and therefore, this was further explored by characterizing the cause of death by pathophysiological process and by organ system. When looking at all female cats, intact cats died at a significantly younger age for all pathophysiological processes except the degenerative, ischemic, metabolic, neoplastic, and toxic causes. Nutritional causes could not be explored as only two cases in total were identified (one female spayed cat and intact male cat). Of note, there were no categories where the median age of death for intact female cats was ≥ the median age of spayed female cats. When looking at female cats ≥1 year of age, in addition to the above-mentioned categories the congenital and inflammatory categories no longer had statistically different ages at the time of death. When evaluating male cats, neutered males also had a survival advantage over intact males in all categories except ischemic, metabolic and neoplastic causes of death. Again, there were no categories where the median age of death was higher for intact males compared to neutered male cats. When looking at male cats ≥1 year of age, in addition to the above-mentioned categories the degenerative, inflammatory and metabolic categories were no longer statistically different. This data is presented in Table 5.

When assessing the cause of death for female cats by organ system, being intact significantly shortened longevity for all organ systems except dermatologic, endocrine and hematopoietic systems. Again, in no organ system category was longevity greater for intact females than spayed females. When looking at female cats ≥1 year of age, the gastrointestinal system, hepatobiliary, musculoskeletal, respiratory and urological systems were no longer statistically different in addition to the systems listed above. Intact males had significantly shorter lifespans for all categories except dermatologic, endocrine, genital, hematopoietic, hepato-biliary and undetermined. The ophthalmic system could not be evaluated as only one neutered male fell into this category. When looking at male cats ≥ 1 year at time of death, the cardiovascular, musculoskeletal and urologic systems were no longer significant. This data is presented in Table 5.

## Discussion

The study reviewed 3,108 cat necropsy examinations and their associated medical records to gain insights on longevity and what factors affect longevity in cats. The study was designed to include only client-owned animals to limit biased results that may be introduced from a broader population; including non-client owned animals would have had limited correlative data, less likely reflecting the population of owned cats who we were interested in studying. Of interest was the finding that the proportion of cats undergoing a necropsy examination has decreased over time. This is similar to what was found in dogs undergoing necropsy examinations at the same institution during much of the time period of the current study [10, 11]. With a declining proportion of cats that die or are euthanized it is possible to miss changes in disease patterns or the emergence of new diseases, although this was beyond the scope of this study. Using a necropsy cohort allows for a more objective and perhaps more accurate diagnosis of disease and cause of death than from clinical data alone, which is why this population was chosen for study [10, 11].

This study found that a greater proportion of intact males lived in an outdoor environment compared to neutered males. This was not true for intact versus spayed female cats.

**Table 5. Cause of death in all cats and cats ≥1 year of age at the time of death categorized by organ system, sex and spay neuter status.**

| Organ System | Sex | All Cats | | | Cats >1 Yr | | |
|---|---|---|---|---|---|---|---|
| | | Number of Cats | Median Age at Death (years) | p value | Number of Cats | Median Age at Death (years) | p value |
| Cardiovascular | F | 9 | 0.40 (IQR 0.12–0.79; Range 0.04–2.88) | <0.0001 | 1 | 2.88 | 0.42 |
| | FS | 90 | 9.59 (IQR 5.09–12.61; Range 0.36–18.94) | | 89 | 9.64 (IQR 5.45–12.61; Range 1.01–18.94) | |
| | M | 15 | 3.72 (IQR 0.79–8.02; Range 0.24–15.18) | 0.005 | 10 | 5.11 (IQR 3.72–8.28; Range 3.42–15.18) | 0.38 |
| | MN | 166 | 8.45 (IQR 4.96–12.13; Range 0.51–21.13) | | 164 | 8.47 (IQR 5.45–12.15; Range 1.33–21.13) | |
| Dermatologic | F | 4 | 7.43 (IQR 2.77–11.75; Range 0.52–13.66) | 0.51 | 3 | 9.84 (IQR 5.01–13.66; Range 5.01–13.66) | 1.00 |
| | FS | 35 | 11.93 (IQR 8.41–13.67; Range 0.51–18.41) | | 33 | 12.17 (IQR 8.94–13.67; Range 3.25–18.41) | |
| | M | 2 | 1.10 (IQR 0.67–1.53; Range 0.67–1.53) | 0.15 | 1 | 1.53 | 0.44 |
| | MN | 32 | 10.04 (IQR 5.68–12.67; Range 2.63–17.76) | | 32 | 10.04 (IQR 5.68–12.67; Range 2.63–17.76) | |
| Endocrine | F | 3 | 14.02 (IQR 1.02–21.15; Range 1.02–21.15) | 1.0000 | 3 | 14.02 (IQR 1.02–21.15; Range 1.02–21.15) | 1.00 |
| | FS | 66 | 13.05 (IQR 9.32–15.18; Range 2.13–21.85) | | 66 | 13.05 (IQR 9.32–15.18; Range 2.13–21.85) | |
| | M | 3 | 9.47 (IQR 9.41–14.51; Range 9.41–14.51) | 1.0000 | 3 | 9.47 (IQR 9.41–14.51; Range 9.41–14.51) | 1.00 |
| | MN | 93 | 12.35 (IQR 9.91–15.01; Range 3.15–19.86) | | 93 | 12.35 (IQR 9.91–15.01; Range 3.15–19.86) | |
| Gastrointestinal | F | 9 | 2.77 (IQR 0.23–9.77; Range 0.08–17.71) | 0.04 | 5 | 9.77 (IQR 3.02–15.06; Range 2.77–17.71) | 1.00 |
| | FS | 139 | 10.85 (IQR 8.02–13.98; Range 0.66–21.24) | | 136 | 11.00 (IQR 8.35–14.09; Range 1.53–21.24) | |
| | M | 15 | 1.21 (IQR 0.21–2.90; Range 0.13–15.87) | <0.0001 | 8 | 2.59 (IQR 1.60–10.23; Range 1.21–15.87) | 0.04 |
| | MN | 135 | 10.65 (IQR 7.64–13.76; Range 0.55–19.85) | | 128 | 10.97 (IQR 8.15–13.80; Range 1.11–19.85) | |
| Genital | F | 6 | 1.61 (IQR 1.05–2.08; Range 0.24–10.36) | 0.005 | 5 | 1.73 (IQR 1.48–2.08; Range 1.05–10.36) | 0.01 |
| | FS | 13 | 12.15 (IQR 9.55–13.58; Range 4.96–19.87) | | 13 | 12.15 (IQR 9.55–13.58; Range 4.96–19.87) | |
| | M | 1 | 6.00 | 1.0000 | 1 | 6.00 | 1.00 |
| | MN | 1 | 9.35 | | 1 | 9.35 | |
| Hematopoetic | F | 6 | 5.93 (IQR 1.27–12.54; Range 0.45–15.03) | 1.0000 | 5 | 10.36 (IQR 1.50–12.54; Range 1.27–15.03) | 1.00 |
| | FS | 32 | 7.28 (IQR 3.05–11.11; Range 0.22–19.90) | | 30 | 7.80 (IQR 4.58–11.22; Range 1.05–19.90) | |
| | M | 9 | 1.64 (IQR 0.82–8.36; Range 0.45–14.99) | 0.16 | 5 | 8.36 (IQR 2.81–10.04; Range 1.64–14.99) | 1.00 |
| | MN | 57 | 7.46 (IQR 3.50–11.08; Range 0.62–18.04) | | 51 | 8.56 (IQR 4.10–11.14; Range 1.06–18.04) | |
| Hepato-billiary | F | 8 | 4.13 (IQR 1.35–6.27; Range 0.02–11.01) | 0.0200 | 6 | 4.45 (IQR 4.07–7.83; Range 2.58–11.01) | 0.16 |
| | FS | 74 | 8.56 (IQR 6.02–12.08; Range 0.88–21.44) | | 73 | 8.60 (IQR 6.02–12.08; Range 1.64–21.44) | |
| | M | 5 | 5.48 (IQR 1.96–13.30; Range 0.36–14.44) | 1.0000 | 4 | 9.39 (IQR 3.72–13.87; Range 1.96–14.44) | 1.00 |
| | MN | 64 | 8.83 (IQR 6.85–11.58; Range 0.51–17.62) | | 63 | 8.86 (IQR 6.88–11.65; Range 1.35–17.62) | |
| Musculoskeletal | F | 6 | 0.29 (IQR 0.17–0.64; Range 0.02–5.20) | 0.0001 | 1 | 5.20 | 0.59 |
| | FS | 38 | 10.12 (IQR 7.56–13.72; Range 0.52–19.39) | | 37 | 10.35 (IQR 7.81–13.72; Range 2.02–19.39) | |
| | M | 9 | 0.53 (IQR 041–2.93; Range 0.01–8.70) | 0.0008 | 4 | 3.53 (IQR 2.36–6.41; Range 1.78–8.70) | 0.13 |
| | MN | 61 | 8.47 (IQR 6.60–12.12; Range 0.74–17.73) | | 59 | 8.99 (IQR 6.71–12.33; Range 1.08–17.73) | |
| Neurological | F | 23 | 1.11 (IQR 0.53–2.69; Range 0.06–16.13) | <0.0001 | 13 | 2.38 (IQR 1.52–4.73; Range 1.06–16.13) | 0.0003 |
| | FS | 127 | 9.18 (IQR 4.37–13.49; Range 0.34–20.49) | | 118 | 9.64 (IQR 5.54–13.78; Range 1.03–20.49) | |
| | M | 15 | 0.91 (IQR 0.43–1.96; Range 0.27–8.90) | <0.0001 | 6 | 2.81 (IQR 1.78–4.80; Range 1.00–8.90) | 0.001 |
| | MN | 159 | 9.04 (IQR 5.40–12.85; Range 0.18–20.66) | | 153 | 9.25 (IQR 5.90–12.96; Range 1.07–20.66) | |
| Ophthalmologic | F | 1 | 0.26 | 0.35 | 0 | | |
| | FS | 4 | 11.45 (IQR 10.23–13.97; Range 9.32–16.18) | | 4 | 11.45 (IQR 10.23–13.97; Range 9.32–16.18) | |
| | M | 0 | | | 0 | | |
| | MN | 1 | 18.07 | | 1 | 18.07 | |
| Respiratory | F | 14 | 2.4 (IQR 0.62–5.35; Range 0.10–13.33) | 0.0003 | 9 | 5.05 (IQR 3.02–11.73; Range 1.24–13.33) | 0.05 |
| | FS | 161 | 10.32 (IQR 7.32–13.41; Range 0.41–20.12) | | 156 | 10.60 (IQR 7.65–13.47; Range 1.58–20.12) | |
| | M | 17 | 0.75 (IQR 0.14–2.24; Range 0.01–8.14) | <0.0001 | 7 | 2.35 (IQR 2.08–7.25; Range 1.19–8.14) | 0.001 |
| | MN | 151 | 10.31 (IQR 6.34–13.33; Range 0.34–17.63) | | 145 | 10.81 (IQR 6.67–13.37; Range 1.11–8.14) | |

(*Continued*)

**Table 5.** (Continued)

| Organ System | Sex | All Cats | | | Cats >1 Yr | | |
|---|---|---|---|---|---|---|---|
| | | Number of Cats | Median Age at Death (years) | p value | Number of Cats | Median Age at Death (years) | p value |
| Multi-organ | F | 49 | 0.69 (IQR 0.48–2.77; Range 0.05–16.24) | <0.0001 | 20 | 3.26 (IQR 2.11–8.30; Range 1.01–16.24) | 0.0016 |
| | FS | 225 | 9.30 (IQR 3.96–13.01; Range 0.28–19.75) | | 204 | 10.01 (IQR 5.88–13.35; Range 1.02–19.75) | |
| | M | 83 | 0.69 (IQR 0.4 1.95; Range 0.11 19.05) | <0.0001 | 29 | 2.66 (IQR 1.57–4.97; Range 1.03–19.05) | 0.0002 |
| | MN | 286 | 7.25 (IQR 2.57–11.95; Range 0.27–20.44) | | 255 | 8.55 (IQR 4.11–12.23; Range 1.02–20.44) | |
| Undetermined | F | 5 | 0.12 (IQR 0.12–0.25; Range 0.01–1.20) | 0.002 | 1 | 1.20 | 0.30 |
| | FS | 18 | 10.24 (IQR 7.09–13.04; Range 4.53–17.47) | | 18 | 10.24 (IQR 7.09–13.04; Range 4.53–17.47) | |
| | M | 6 | 0.12 (IQR 0.12–0.18; Range 0.09–15.30) | 0.06 | 1 | 15.30 | 0.57 |
| | MN | 11 | 10.19 (IQR 6.47–12.28; Range 4.02–19.70) | | 11 | 10.19 (IQR 6.47–12.28; Range 4.02–19.70) | |
| Urological | F | 19 | 2.60 (IQR 0.75–12.40; Range 0.22–17.33) | 0.001 | 13 | 9.56 (IQR 2.60–13.82; Range 1.39–17.33) | 0.44 |
| | FS | 173 | 11.58 (IQR 6.53–15.16; Range 0.71–21.75) | | 170 | 11.65 (IQR 6.69–15.28; Range 1.12–21.75) | |
| | M | 12 | 0.92 (IQR 0.24–5.74; Range 0.11–13.84) | 0.001 | 6 | 5.74 (IQR 1.80–8.93; Range 1.13–13.84) | 0.38 |
| | MN | 207 | 8.74 (IQR 5.88–12.63; Range 0.24–21.19) | | 199 | 9.07 (IQR 6.12–12. 76; Range 1.02–21.19) | |

Interestingly the proportion of FeLV positive cats was higher among intact males than among castrated males, with no significant differences in FeLV status found between intact and spayed females. This could be related to the increased exposure risk from living in an outdoor environment as well as behavioral differences between neutered and intact males. In contrast, FIV status and differences in environmental housing did not have a statistically significant relationship for either sex. The longer lifespan of cats who were FIV positive compared to those who were FeLV positive together with the possibility that cats known to be FIV positive may have been more likely to be kept indoors may account for this.

When looking at causes of death and morbidity based on histological findings it is important to realize that even though considered a gold standard histopathology may not be able to precisely determine a cause of death. For example, in inflammatory diseases confidence in how much a mild or moderate inflammatory response leads to a particular disease process or co-morbidity can be difficult to determine in most organ systems. The most common organ system found to be the cause of death was the multiorgan/systemic category. This was followed by the urological system. The most common pathophysiologic cause of death was cancer followed by infectious disease. When looking at classification of disease, the most common cause of death was cancer followed by renal disease, feline infectious peritonitis and cardiac disease. These four causes of death accounted for 58.52% percent of all deaths indicating the importance of these four disease categories on feline mortality and longevity and highlighting areas of future research that need to be performed to impact both quality and quantity of life.

Cancer was identified in 41.3% of cats on their necropsy examinations, with 86.75% dying of their cancer. Interestingly we found that while the majority of cats had a single cancer identified (87.45%), 2 to 5 distinct cancers were noted in the remaining cats. The most common category of cancer was round cell tumors.

Renal disease was also found to have a significant contribution to both morbidity and mortality; 62.84% of cats had some form of renal pathology identified with 20.79% of these cases having their death attributed to renal disease. The majority of these cats were diagnosed with tubular/interstitial disease with nearly 39% of the cats in this study having histopathological evidence of this disease process, which is consistent with other reports [13]. Again here it is important to note that while the majority of deaths were not directly attributable to their renal disease it is difficult to say if they contributed to either loss of quality of life or other co-morbidities that may have limited their longevity.

Cardiac disease also had a significant impact on cats, with 27.90% of cats found to have some form of cardiac pathology and 25.84% of these cats dying from it. Hypertrophic cardiomyopathy in cats has been evaluated in a large multicenter study, which included this institution. That study found cardiovascular-related death in 27.9% of 1008 cats diagnosed with this condition [6].

In a study looking at causes of death in cats living in free roaming colonies in the city of Milan, Italy the most common causes of death included inflammatory diseases (which included infectious causes), organ failure and trauma [14]. It is not surprising that the results in this study differed given that these animals were unowned and faced different pressures than owned cats. Feline infectious peritonitis was still prominent cause of death in these cats with 13/186 (7%) cats dying of this disease, which was quite similar to the 6.72% of cats in our study who died of this.

The median age of death was 9.07 years among all cats and 9.92 years among cats who lived to ≥ 1 year of age. To better control for the effect that death at a young age might introduce, statistical analysis was performed for both the entire group of cats and for those ≥ 1 year of age. One year was chosen as the age for a cat to become an adult based on several studies that used this age; however, we acknowledge there is no real defined age for when a kitten becomes an adult cat [15, 16]. The longevity found in our study is shorter than what another study reported for cats in England, which might relate to this being a necropsy-based population or that of a referral institution [9].

To try to determine what environmental and demographic factors affected a cat's longevity, regression analysis was performed. Being FeLV positive decreased age at death (p<0.001), being intact, male or female combined, decreased age at death (p<0.0001), while being a spayed female (p<0.0001) and being a neutered male (p<0.0001) increased longevity significantly. Environmental housing status, FIV status, or being purebred versus mixed breed did not have significant effects on the model.

We found that intact animals had a decreased longevity when compared to spayed or neutered animals. This held true across the board for all organ systems and pathophysiologic processes, although not all of these differences were found to be statistically different. Interestingly, a study by Hamilton et al. from 1969 that examined cats from a different university teaching hospital found that being intact negatively impacted longevity for both male and female cats [17].

In most mammalian species females on average live longer than males [18, 19], which was also found in this study. While the exact causes for this are not known, it has been linked to epigenetic changes. Interestingly, Sugrue et al. found that castrated male sheep had decelerated aging compared to their intact counterparts. This was at least in part due to the removal of androgens [20]. This could help explain the differences in longevity seen between intact and neutered male cats but does not explain the differences in longevity seen between intact and spayed female cats.

We were unable to determine the age at time of spay for female cats and/or if they had any litters, as this information was not explicitly recorded in the majority of records. There is mixed evidence in humans that reproduction, particularly at a younger age, decreases longevity, but this could not be correlated in cats since this information was not available for most cases [21].

One limitation of this study is that the cases come from a veterinary medical teaching hospital collection, with a majority of cases being referral or emergency in nature. This may have selected for a less healthy population with a shorter longevity than might have been found in the general population of owned cats, such as those presenting to primary care practices [9]. By using cases from a veterinary medical teaching hospital with a referral base bias may have

been introduced because of an inconsistent catchment area, referral of patients thought to have a better prognosis, or referral of those cats owned by people of a higher economic status. Other limitations include missing or incomplete data for several factors including the age and environmental data for some cats.

## Conclusions

This study determined longevity in a population of cats undergoing a necropsy examination at a veterinary medical teaching hospital over a period of 30 years. The most common disease process found was cancer. Renal disease also was found in a large portion of the cases. As other studies have also found, intact males and females had a significantly shorter longevity than their castrated or spayed counterparts.

## Supporting information

**S1 Fig. Proportion of cats undergoing necropsy examination by year.** Graph showing the proportion of cats undergoing necropsy by year from 1989–2019. In addition, the regression line showing a decrease in the proportion of cats undergoing a necropsy along with the 95% confidence intervals represented by the shaded area is shown.
(TIF)

**S1 Table. Table showing breeds of cats in the study along with their ages at time of death.**
(DOCX)

**S2 Table. Table with the main cause of death of client owned cats in this study.**
(DOCX)

**S1 Dataset. The entire dataset of cats analyzed in this study.** This file contains the dataset used for the study. A code book is available at the bottom of the file.
(XLSX)

## Author Contributions

**Conceptualization:** Michael S. Kent, Robert Rebhun.

**Data curation:** Michael S. Kent, Sophie Karchemskiy, William T. N. Culp, Patricia A. Pesavento, Robert Rebhun.

**Formal analysis:** Michael S. Kent, William T. N. Culp, Amandine T. Lejeune, Patricia A. Pesavento, Christine Toedebusch, Rachel Brady, Robert Rebhun.

**Funding acquisition:** Michael S. Kent.

**Investigation:** Amandine T. Lejeune, Christine Toedebusch.

**Methodology:** Michael S. Kent, Sophie Karchemskiy, Robert Rebhun.

**Writing – original draft:** Michael S. Kent, Amandine T. Lejeune, Robert Rebhun.

**Writing – review & editing:** Michael S. Kent, Sophie Karchemskiy, William T. N. Culp, Patricia A. Pesavento, Christine Toedebusch, Rachel Brady, Robert Rebhun.

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
