## [Decision Letter · Decision Letter 0]

26 Sep 2022

PONE-D-22-19408Longevity and mortality in cats: A single institution necropsy study of 3108 cases (1989-2019)PLOS ONE

Dear Dr. Kent,

Thank you for submitting your manuscript to PLOS ONE. After careful consideration, we feel that it has merit but does not fully meet PLOS ONE’s publication criteria as it currently stands. Therefore, we invite you to submit a revised version of the manuscript that addresses the points raised during the review process.

We look forward to receiving your revised manuscript.

Kind regards,

Silvia Sabattini

Academic Editor

PLOS ONE

Journal Requirements:

2. We understand that the electronic medical record database of the UC Davis William R. Pritchard Veterinary Medical Teaching Hospital was accessed to evaluate the records of all cats undergoing necropsy examinations. Please clarify whether the authors had access to personal identifying information from the cat owners.

5. Please upload a copy of Supplemental Table 1 and Table 2 which you refer to in your text on page 10 and 16.

Reviewers' comments:

Reviewer's Responses to Questions

**Comments to the Author**

1. Is the manuscript technically sound, and do the data support the conclusions?

Reviewer #1: Yes

Reviewer #2: Yes

2. Has the statistical analysis been performed appropriately and rigorously? 

Reviewer #1: Yes

Reviewer #2: Yes

3. Have the authors made all data underlying the findings in their manuscript fully available?

Reviewer #1: Yes

Reviewer #2: Yes

4. Is the manuscript presented in an intelligible fashion and written in standard English?

Reviewer #1: Yes

Reviewer #2: Yes

5. Review Comments to the Author

Reviewer #1: The authors did an outstanding epidemiological study on the main causes of death in cats, reviewing more than 3000 necropsies in detail, and putting the data in correct correlation and statistics. This paper deserves publication as it’s important the scientific contribution given to the veterinary community.

Just very few corrections to the manuscript are reported.

Line 96 – There’s a typo in the word “Research”.

Line 96 – “With a complete” instead of “completed”.

Line 273 – Typo

Line 297 – Please rephrase the period.

Line 318 – is “at” a typo?

Line 334 – What kind of macroscopic or histopathological findings were present in cardiac disease bearing animals? Could you please rephrase the entire period, because it doesn’t sound fluid in reading. Thanks.

Line 510-513 – This was not mentioned in the results (as previous comment requested). Can you please add more specific details of the cardiac pathologies found?

Table 3 and 4 - These tables are really difficult to understand. My suggestion would be to make them in a clear way, or delete them at all.

Reviewer #2: The manuscript is very good and the topic very interesting.

The manuscript is well written, the study is well designed and based on several well exposed exposed data. Results are very interesting and well commented in the discussion.

The manuscript is a little bit long, but I suggest to maintain this lenght because the manuscript is not boring: it is simply very complete.

It is maybe the first time in my life that I really have no critics to do.

I have a simple suggestion to improve the study: practically no data are present in literature concerning causes of death in pet animals. A recent paper, published on "Animals", analysed the causes of death in colony cats. It would be interesting to insert, in the present manuscript, few sentences comapring the causes of death of owned cat with those of free cats

6. PLOS authors have the option to publish the peer review history of their article (what does this mean?). If published, this will include your full peer review and any attached files.

Reviewer #1: No

Reviewer #2: No

---

## [Author Response · Author response to Decision Letter 0]

5 Oct 2022

Dear editor and reviewers;

Thank you very much for the effort and time put into reviewing our manuscript. We have done our best to address the issues raised and look forward to your re-review of the manuscript.

Please find answers to all the comment integrated into the text below.

Respectfully submitted,

Michael Kent, DVM, DACVIM (Oncology), DACVR (Radiation Oncology), ECVDI (RO-Add on)

I have reviewed the files and believe that I have followed the guidelines for this.

2. We understand that the electronic medical record database of the UC Davis William R. Pritchard Veterinary Medical Teaching Hospital was accessed to evaluate the records of all cats undergoing necropsy examinations. Please clarify whether the authors had access to personal identifying information from the cat owners.

As the faculty and students at our institution we have access to all records and all information that is in the EMR, although owner information beyond contact information is not stored in our EMR. For the purposes of this study this limited owner information that is collected was not accessed. All owner’s given verbal or written consent for necropsy examination and are informed that the results are stored for research as our tissue samples (which were not accessed for this study).

These were the same funder but I have expanded this to include all the information in the financial disclosure section. 

We have made a new supplemental file with the data set used in this study to ensure full access to the data.

5. Please upload a copy of Supplemental Table 1 and Table 2 which you refer to in your text on page 10 and 16.

These have been uploaded along with the revised manuscript.

These have been placed at the end after the reference list.

The reference list has been reviewed. No paper that we have cited has been retracted.

Reviewer #1: The authors did an outstanding epidemiological study on the main causes of death in cats, reviewing more than 3000 necropsies in detail, and putting the data in correct correlation and statistics. This paper deserves publication as it’s important the scientific contribution given to the veterinary community.

We appreciate your opinion on our work and its value. Thank you.

Just very few corrections to the manuscript are reported.

Line 96 – There’s a typo in the word “Research”.

Corrected – thank you

Line 96 – “With a complete” instead of “completed”.

Change made as requested.

Line 273 – Typo

Change made as requested.

Line 297 – Please rephrase the period.

Rephrased.

Line 318 – is “at” a typo?

Yes it was – thank you for finding this.

Line 334 – What kind of macroscopic or histopathological findings were present in cardiac disease bearing animals? Could you please rephrase the entire period, because it doesn’t sound fluid in reading. Thanks.

We have edited the sentence as suggested and added the following to the manuscript in the results section.

A total of 867 (27.90%) cats were found to have some form of cardiac pathology with it being attributed as the cause of death in 224 (25.84%) of those cases. The most common cardiac abnormality noted was hypertrophic cardiomyopathy (392 cases with it being identified as the cause of death in 145 cases). This was followed by endocardiosis with 85 cases (where it was identified as the main driver of mortality in 13 cases), myocarditis with 59 cases (where it cause death in 8 cases), and myocardial fibrosis (where it was identified as the cause of death in 8 cases). 

Line 510-513 – This was not mentioned in the results (as previous comment requested). Can you please add more specific details of the cardiac pathologies found?

Please see above comment where this information was added.

Table 3 and 4 - These tables are really difficult to understand. My suggestion would be to make them in a clear way, or delete them at all.

We have tried to better explain the table but do feel the information is valuable for determining how often multiple cancers or types of renal disease were present. We have made the following changes to the table lead in and titles to help better explain the data in the tables:

A complete table of tumor categories broken down by the number of cancers for each cat is available in Table 3.

Table 3: Frequency of occurrence of tumor types and number of tumors seen in each cat.

For a complete breakdown of kidney disease in these cats see Table 4. This table also shows which cats had multiple renal pathologies and which type of kidney disease occurred together in a single cat.

Reviewer #2: The manuscript is very good and the topic very interesting.

The manuscript is well written, the study is well designed and based on several well exposed exposed data. Results are very interesting and well commented in the discussion.

The manuscript is a little bit long, but I suggest to maintain this length because the manuscript is not boring: it is simply very complete.

It is maybe the first time in my life that I really have no critics to do.

Thank you for your review of our work and we value and appreciate your opinion. 

I have a simple suggestion to improve the study: practically no data are present in literature concerning causes of death in pet animals. A recent paper, published on "Animals", analysed the causes of death in colony cats. It would be interesting to insert, in the present manuscript, few sentences comparing the causes of death of owned cat with those of free cats

We were unaware of this study – thank you for bringing it to our attention. We have added the following to the manuscript in the discussion section:

In a study looking at causes of death in cats living in free roaming colonies in the city of Milan, Italy the most common causes of death included inflammatory diseases (which included infectious causes), organ failure and trauma[14]. It is not surprising that the results in this study differed given that these animals were unowned and faced different pressures than owned cats. Feline infectious peritonitis was still prominent cause of death in these cats with 13/186 (7%) cats dying of this disease, which was quite similar to the 6.72% of cats in our study who died of this.

---

## [Decision Letter · Decision Letter 1]

14 Nov 2022

Longevity and mortality in cats: A single institution necropsy study of 3108 cases (1989-2019)

PONE-D-22-19408R1

Dear Dr. Kent,

We’re pleased to inform you that your manuscript has been judged scientifically suitable for publication and will be formally accepted for publication once it meets all outstanding technical requirements.

Kind regards,

Silvia Sabattini

Academic Editor

PLOS ONE

Additional Editor Comments (optional):

Reviewers' comments:

Reviewer's Responses to Questions

**Comments to the Author**

1. If the authors have adequately addressed your comments raised in a previous round of review and you feel that this manuscript is now acceptable for publication, you may indicate that here to bypass the “Comments to the Author” section, enter your conflict of interest statement in the “Confidential to Editor” section, and submit your "Accept" recommendation.

Reviewer #1: All comments have been addressed

Reviewer #2: All comments have been addressed

2. Is the manuscript technically sound, and do the data support the conclusions?

Reviewer #1: Yes

Reviewer #2: Yes

3. Has the statistical analysis been performed appropriately and rigorously? 

Reviewer #1: Yes

Reviewer #2: Yes

4. Have the authors made all data underlying the findings in their manuscript fully available?

Reviewer #1: Yes

Reviewer #2: (No Response)

5. Is the manuscript presented in an intelligible fashion and written in standard English?

Reviewer #1: Yes

Reviewer #2: Yes

6. Review Comments to the Author

Reviewer #1: (No Response)

Reviewer #2: Authors accepted all comments and did related corrections

The manuscript is now improved and, on my opinion, suitable for publication

7. PLOS authors have the option to publish the peer review history of their article (what does this mean?). If published, this will include your full peer review and any attached files.

Reviewer #1: No

Reviewer #2: No

---

## [Editor Report · Acceptance letter]

21 Nov 2022

PONE-D-22-19408R1 

Longevity and mortality in cats: A single institution necropsy study of 3108 cases (1989-2019) 

Dear Dr. Kent:

I'm pleased to inform you that your manuscript has been deemed suitable for publication in PLOS ONE. Congratulations! Your manuscript is now with our production department. 

Kind regards, 

on behalf of

Dr. Silvia Sabattini 

Academic Editor

PLOS ONE